# Menthol Mouth Rinsing Maintains Relative Power Production during Three-Minute Maximal Cycling Performance in the Heat Compared to Cold Water and Placebo Rinsing

**DOI:** 10.3390/ijerph19063527

**Published:** 2022-03-16

**Authors:** Seana Crosby, Anna Butcher, Kerin McDonald, Nicolas Berger, Petrus J. Bekker, Russ Best

**Affiliations:** 1Centre for Sport Science & Human Performance, Waikato Institute of Technology, Hamilton 3200, New Zealand; seanakcrosby@gmail.com (S.C.); annajbutcher@gmail.com (A.B.); kerin.mcdonald@wintec.ac.nz (K.M.); jako.bekker@wintec.ac.nz (P.J.B.); 2School of Health and Life Sciences, Teesside University, Middlesbrough TS1 3BX, UK; n.berger@tees.ac.uk

**Keywords:** menthol, cycling, maximal exercise, heat

## Abstract

Previous menthol studies have demonstrated ergogenic effects in endurance-based activity. However, there is a need for research in sports whose physiological requirements exceed maximal aerobic capacity. This study assessed the effects of 0.1% menthol mouth-rinsing upon a modified three-minute maximal test in the heat (33.0 ± 3.0 °C; RH 46.0 ± 5.0%). In a randomised crossover single blind placebo-controlled study, 11 participants completed three modified maximal tests, where each trial included a different mouth rinse: either menthol (MEN), cold water (WAT) or placebo (PLA). Participants were asked to rate their thermal comfort (TC), thermal sensation (TS) and rating of perceived exertion (RPE) throughout the test. Heart rate, core temperature, oxygen uptake (VO_2_), ventilation (VE) and respiratory exchange ratio (RER) were monitored continuously throughout the test, alongside cycling power variables (W; W/kg). A blood lactate (BLa) level was taken pre- and post- test. Small to moderate effects (Cohen’s *d* and accompanying 90% confidence intervals) between solutions MEN, WAT and PLA were observed towards the end of the test in relation to relative power. Specifically, from 75–105 s between solutions MEN and WAT (ES: 0.795; 90% CI: 0.204 to 1.352) and MEN and PLA (ES: 1.059; 90% CI: 0.412 to 1.666), this continued between MEN and WAT (ES: 0.729; 90% CI: 0.152 to 1.276) and MEN and PLA (ES: 0.791; 90% CI: 0.202 to 1.348) from 105–135 s. Between 135–165 s there was a moderate difference between solutions MEN and WAT (ES: 1.058; 90% CI: 0.411 to 1.665). This indicates participants produced higher relative power for longer durations with the addition of the menthol mouth rinse, compared to cold water or placebo. The use of menthol (0.1%) as a mouth rinse showed small performance benefits for short duration high intensity exercise in the heat.

## 1. Introduction

Menthol is an organic compound with the ability to impart non-thermal cooling effects on targeted receptors, across body regions. Menthol is utilised in many commercial products for its perceptual cooling effect and is nontoxic for human consumption when administered at commercial doses [1,2]. Perceptual cooling via menthol is an extension of thermal cooling for athletes performing exercise in the heat. Original thermal cooling modalities included ice slurry ingestion, cold air exposure and cold-water immersions, which typically lead to a physiological reduction in core temperature (T_core_) [3,4].

The recent combination of menthol with other thermal cooling methods has sparked interest in both research and applied settings as a novel and safe ergogenic aid, when administered in accordance with the scientific literature [1]. It is important to acknowledge that although menthol does not prevent heat gain or reduce T_core_, it aids in the athletes feeling perceptually cool and being able to perform for a longer period of time [5,6,7] or at a higher power output [8,9,10]. As long as human maximum temperature is not reached and other heat illness factors are minimised [1,11], improvements in thermal comfort have been shown to enhance performance independent of an athlete’s thermal state [12,13].

There have been multiple studies that have assessed endurance exercise performance in the heat alongside menthol administration [5,14,15]. Trials that consist of exercise protocols exceeding ten minutes in duration have seen positive effects of menthol use through both topical and oral applications [1]. Topical application of menthol preceded studies on rinsing or ingestion. Gillis et al. [14], were among the first studies to use menthol in a spray form onto the skin, using menthol concentrations of 0.01% and 0.2%, in a spray form applied to the participant’s body five minutes prior to the trial starting. Using menthol as a mouth rinse followed on from the topical spray, with similar concentrations examined. Mouth rinsing may induce a more rapid and intense experience of menthol and its cooling properties, due to decreased thickness of the stratum corneum and potential for increased stimulation of transient receptor potential melastatin 8 (TRPM8) receptors and trigeminal nerves via the oral cavity [16,17,18], lower concentrations may be preferred by some athletes, so as to minimize any potential irritant effects.

Cold water also warrants consideration as a simple and effective rinse during exercise. Its temperature will also stimulate TRPM8 and trigeminal pathways, as a result of a physiological reduction in local temperature within the oral cavity. Ingestion of cool/cold beverages are preferred during exercise compared to room temperature or warm beverages [19,20,21,22], and are deemed to be more palatable, thus promoting further ingestion or satisfying thirst to a greater extent. The particular sensation that is imparted by ‘cold’ is that of being of refreshed [23,24,25]. This has the potential to decrease the rating of perceived exertion or improve task motivation [20], along with potentially modifying sensations of airflow or nasal patency [26,27]. If sufficient fluid volumes are also ingested, then a reduction in core temperature may also contribute to improvements in performance [19,20], however ice ingestion is considered a more time-efficient means of achieving a desired reduction in core temperature, and indeed when menthol is co-ingested with a liquid, has been shown to increase its ergogenicity inversely proportional to the beverage temperature [10]. Taking the above into account the inclusion of cold-water swills or ingestion in menthol trials is warranted due to their common stimulatory pathways, but different physiological outcomes.

As mentioned, similar concentrations of menthol to those administered topically have been used as a mouth rinse, with 0.01% the most commonly applied to date. Studies have typically shown that the use of menthol during endurance-based activities has resulted in a smaller effect on thermal perception compared to other cooling methods such as ice slurry and cold-water immersion or application [2,17,28]. Menthol being applied as a mouth rinse targets a smaller thermally sensitive area compared to a topical spray, or other more aggressive cooling methodologies that elicit a reduction in core body temperature. However, a greater menthol concentration could potentially eliminate variability due to individuals’ differing menthol sensitivity thresholds [14,29,30,31]. It is important to note that at higher concentrations menthol acts as an irritant, but when administered at appropriate concentrations and time intervals, irritation and desensitization are mitigated [32,33]. This indicates that the timing of menthol administration relative to, or within, an exercise bout is a pertinent consideration for practitioners and athletes.

Considering current evidence, a recent consensus statement found insufficient research regarding the effect of menthol on either intermittent or explosive exercise, with literature exploring the effect of menthol on short duration high-intensity bouts of exercise also scarce [1,34]. Previously, longer durations have been examined, with participants completing trials to exhaustion or extended time trials that rely on aerobic energy provision [5,14,35]. When exercising in the heat, these durations permit a sufficient ‘hyperthermic dose’, with T_core_ significantly elevated as a correlate of exercise duration. More specifically, as exercise duration in the heat increases, so too does the cardiovascular, thermoregulatory and ventilatory strain experienced by an athlete [36,37]. The time-course of these physiological responses presents a seemingly ideal metabolic milieu for menthol to exert its primary perceptual cooling effects and its secondary physiological effects, such as increasing ventilation (VE) [15]. When exercising in the heat for a short duration at near maximal intensity, the hyper-thermic potential of the exercise bout is limited, but some physiological responses akin to when athletes experience hyperthermia may still occur, due to the intensity of the exercise bout. These may include but are not limited to increases in VE, respiratory exchange ratio and carbohydrate oxidation [38,39], effectively increasing the percentage of maximal oxygen uptake (VO_2max_) assigned to the task.

The three-minute all-out test [40,41] was designed to elicit maximal oxygen uptake, and profile the rate of decay to critical power—capturing the asymptote of the severe intensity domain. The three-minute duration is important, as this appears to be the duration above (and the intensity below) which athletes demonstrate an increased aerobic contribution to energy production [38,39,42]. Using the three-minute duration also simulates competitive events such as the individual or team pursuit, and is mirrored in events such as 1500 m running races, where the contribution from aerobic energetic pathways are maximal (i.e., ≥VO_2max_), but also require a notable energetic contribution from glycolytic sources too [38,39,42], event tactical demands notwithstanding. Given the previous work in prolonged exercise tasks and the limited evidence in short duration efforts following menthol rinsing, the three-minute test is a natural progression in the line of inquiry; as it assesses short duration and maximal effort exercise simultaneously, which have only been investigated following menthol previously to a limited extent, employing repeated sprint designs [34,43]. Based on the rationale presented above, the aim of the present study was to examine whether a single menthol mouth rinse influenced performance of and/or thermal perception during a three-minute maximal test compared to cold water and placebo mouth rinses under hot environmental conditions.

## 2. Materials and Methods

Ethical approval for this study was obtained from the Waikato Institute of Technology’s Human Ethics Research Group (Approval code: WTFE13130820). Participants gave verbal and written consent upon arrival at the laboratory prior to participation and had the opportunity to ask questions about study design, intensity etc. to ensure informed consent was maintained.

### 2.1. Participants

Eleven healthy non-heat-acclimated male (*n* = 6) and female (*n* = 5) participants (height: 171.2 ± 11.0 cm; mass: 75 ± 13 kg; age: 25 ± 5 years) were included in this study. To ensure non-heat-acclimated participation, research was conducted over autumn and winter months, with no participant travelling to hot (≥28 °C) climate for ≥two months, prior to study commencement [28,31]. Inclusion criteria required participants to have general fitness ability, with prior experience in exercise testing procedures and being self-reported as free from injury. Participants cycled three to seven times per week, for either dedicated exercise, competitive or cross-training purposes. All participants were familiar with the use of a WattBike™ at self-reported maximal intensity. Participants were instructed to avoid non-habitual consumption of alcohol or caffeinated products 24 h prior to each testing session, as well as abstain from vigorous exercise 24 h before sessions. Participants reported to the laboratory to complete introductory measures that included, height, body mass and health questionnaires.

### 2.2. General Design

All tests were performed on a WattBike™ Pro cycle ergometer (Watt Bike Ltd., Nottingham, UK). During the first visit participants completed a VO_2max_ test, before completing a familiarisation session and three experimental sessions in the heat chamber. Participants undertook five visits in total.

The VO_2max_ test protocol was completed in temperate conditions (20 °C; 30% humidity). Participants completed a self-selected 10 min warm up before starting the test. The same warm up was then repeated prior to experimental sessions. Participants started the incremental ramp test cycling at 100 W, this increased by 25 W every minute [44]. RPE was measured at the end of each one-minute stage by the participant pointing to a CR-10 RPE scale [45] attached to a board in front of the WattBike™. Oxygen uptake was continuously measured, breath-by-breath, using a Metalyzer CPET device (Coretex Medical, Leipzig, Germany). Maximal oxygen uptake was considered to have occurred at the point of VO_2_ not increasing despite an increase in VE, as per guidelines from Taylor [46] with secondary criteria of a RER > 1.10, high blood lactate levels or RPE approaching or at maximal levels [47]. Blood sampling for lactate analysis was not taken during the first visit. If these criteria were not met, but participants still reached voluntary exhaustion, then the measure was deemed to be VO_2peak_.

During the second visit a full familiarisation session was conducted in which the participants experienced the testing conditions in an environmental heat chamber (33.0 ± 3.0° with RH 46.0 ± 5.0%), where the experimental tests were going to take place. This session included instructions on how to perform the three-minute all-out maximal test, as well an introduction to the thermal perception scales that were used in the testing sessions, experience with potential heat stress and advice on how to recognise heat illnesses throughout the trial. WattBike™ geometry measurements were also established during this session and maintained throughout the study. All participants were asked to raise questions or queries they might have had with the experimental testing and its conditions at this time, but still retained their right to withdraw at any time.

Participants then performed three randomised experimental trials on visits 3–5, which were separated by at least 48 h, to permit adequate recovery. Each of the trials were performed at the same time of the day to minimise potential circadian variation. A general experimental schematic is presented in Figure 1.

### 2.3. Experimental Procedure

Having abstained from vigorous exercise >24 h prior to the tests, all participants were asked to ensure they were well hydrated before arriving at the laboratory. Upon arrival at the laboratory, participants were required to provide a urine sample for a urine specific gravity test (USG) to check their hydration status. If participants USG was ≥1.030, they were asked to ingest 200–500 mL of water before entering the heat chamber [48]. Participants were also advised to consume a habitual diet to minimise the difference in muscle glycogen levels and respiratory exchange ratio (RER) in between visits.

Participants were provided with a rectal thermistor, which was self-inserted 10 cm beyond the anal sphincter. Participants’ body masses were assessed prior to entering the heat chamber and again on exit from the heat chamber to gauge sweat loss for rehydration purposes before ingesting any fluid. When in the heat chamber, the rectal thermistor was connected to a data logger (SQ2010, Grant Instruments, Cambridge, UK) which recorded rectal temperature (T_rec_) throughout the duration of the test(s). Whilst it is acknowledged that the short testing duration was unlikely to elicit hyperthermia, in line with institutional safety requirements, internal temperature was closely monitored while in the heat chamber to ensure prevention of heat induced illnesses.

Upon entering the heat chamber (33.0 ± 3.0 °C with RH 46.0 ± 5.0%), participants were asked to make themselves comfortable on the WattBike™ prior to the commencement of their 10-min self-selected warm-up. Participants wore exercise clothing in which they felt most comfortable; this was shorts or exercise tights, lightweight t-shirt and shoes. The estimated insulative values of environment appropriate sportswear for males is 0.45–0.61 clo and 0.57 clo for females [30,31,49]. Once settled on the bike prior to warm up, resting blood lactate was taken via a fingertip sample (Lactate Pro 2, Arkray Ltd., Koka-shi, Shiga, Japan). Heart rate (Polar RS400, Polar Electro Oy, Kempele, Finland), T_rec_, RPE, thermal comfort (TC) and thermal sensation (TS) were also recorded, following the mouth rinse and fitting of the gas collection mask. VO_2_, VE and RER were then recorded continuously for the remaining duration of the protocol.

The mouth rinse was performed approximately one minute prior to the completion of a modified three-minute all-out maximal test, for which the protocol was obtained via the Watt Bike™ online manual (“The 3 Minute Aerobic Test”, 2021). Participants were asked to cycle ‘all out’ for three minutes. No visual or verbal feedback was provided throughout the test and the participants were blinded to the Wattbike™ screen, so data did not act as an external cue. At the 90 s mark and immediately following the test participants were asked to state their measure of TC, TS and RPE. Physiological respiratory measures were recorded continuously throughout the exercise period and into participant recovery, unless otherwise stated. Figure 1 provides an overview of measurements and measurement intervals.

#### 2.3.1. Physiological Measures

During the trials, a Metalyzer 3B CPET device (Coretex Medical, Leipzig, Germany), was used to analyse expired air throughout the duration of the warmup and the three-minute maximal test. Measures recorded were absolute and relative oxygen uptake (VO_2_; L/min and mL/kg/min), minute ventilation (VE; L/min) and respiratory exchange ratio (RER). Heart rate was recorded via a chest strap (Polar H10, Polar Electro Oy, Kempele Finland) using the Polar team app on a paired iPad (Apple, Los Altos, CA, USA). The rectal thermistor was connected to a data logger (SQ2010, Grant Instruments, Cambridge, UK) with values being recorded pre and post warm up as well as post mouth rinse, halfway through the maximal test and twice during the 5 min recovery period. Blood lactate was taken using a lancet to pin prick the participant’s chosen finger to extract a capillary sample. The blood was collected using a lactate strip attached to a Lactate Pro 2 (Arkray Lactate Pro 2, Minneapolis, MN, USA) [50]. Blood sampling occurred prior to warm up, post mouth-rinse, immediately after the test and 3 min post-completion, during participants’ recovery.

#### 2.3.2. Perceptual Measures

Thermal comfort was measured on a scale from 4 to −4. Polarities equating to 4 being very comfortable and −4 being very uncomfortable [51]. Measurement of thermal sensation was performed using an adapted scale from Young et al. [52], which ranged from 3 to 8. Descriptors aligned with their 0–8 scale, 3 being described as cool and 8 being extremely hot. Conceptual differences between thermal comfort and thermal sensation were described to participants as part of familiarisation. Measures of thermal perception were taken pre and post warmup, post mouth rinse, 90 and 180 s into the 3-min test and 3 min into the participants’ recovery. Importantly, we did not ask participants to respond verbally, but demonstrably, as they simply pointed their finger at the appropriate point on each scale in quick succession. This minimised distraction during the test and was also achievable following maximal effort. Rating of perceived exertion (RPE) was measured at the same time points using Borg’s CR-10 scale [45]. The scale starts at 0 representing no exertion and concludes at 11, representing maximal exertion.

#### 2.3.3. Solution Formulation

A three-way Latin square design was used to assign the participants their trial order ensuring each testing sequence was randomly assigned. Participants were given 25 mL of a solution which was either a placebo (PLA) mouth rinse made up of 4 mL of synthetic non-menthol containing peppermint extract (Hansells, Auckland, New Zealand), cold water (4 °C; WAT), or a menthol concentration of 0.1% mixed with de-ionized water, prepared as per Best et al. ([53]; MEN). All solutions were coloured with green food colouring to encourage participant blinding and to ensure consistent sensory expectancy between solutions. Swilling of the mouth rinse commenced before the trial started and lasted ~5 s before expectoration into a bucket. Participants then commenced the exercise bout. The use of a double-blind protocol was rejected at the ethical approval stage. Given that participants were exercising in the heat, maximally, it was deemed that the researcher, who was present in the chamber with participants, must know what substance was being administered so in the unlikely case of an adverse reaction to menthol, appropriate safety procedures could be implemented.

### 2.4. Statistical Analysis

Statistical analyses were performed using JASP (JASP team (2020) v0.14.1; University of Amsterdam), with statistical significance set a priori at *p* ≤ 0.05. Two-way repeated measures analyses of variance (ANOVA) were used to assess the effects of solution, time and solution × time interactions, following assessment for normal distribution of data according to previously established criteria [54,55]. Where sphericity could not be assumed a Greenhouse–Geisser correction was applied, differences in main effects were further analysed through pairwise comparisons, standardised mean differences (Cohen’s *d*) and accompanying 90% confidence intervals, to express the uncertainty within the estimate. Effect sizes were interpreted as per the recommendations outlined by Hopkins et al. [56], where *small, moderate* and *large* effects were between 0.2–0.6 SD, 0.6–1.2 SD and >1.2 SD, respectively. Data for power output and respiratory variables are presented in 30 s increments, commencing at 15 s and concluding at 165 s (i.e., the middle 150 s), this minimises variability due to the removal of time associated with initial acceleration and (fatigue as a result of) an end-spurt effort [41,57,58].

Changes in performance were considered practically meaningful if the mean response exceeded the variability of the test. Performance variability is presented in two ways: a more conservative fixed approach, and a less conservative funnel. The initial approach considers variability, and thus the threshold for improved performance, as a flat line fixed at 12 W, adapted from Johnson et al., who suggested typical error of 11.1 W for 150 s power output [57] for the duration of the test. The second approach, funnels down from the initial 12 W value at the 15–45 s interval to 5 W at the final time interval (135–165 s), in accordance with previous work that has assessed the reliability of end power following a three-minute all-out test [58].

## 3. Results

All data sets were normally distributed in accordance with assumption checks.

### 3.1. Performance 

Absolute power performance was not significantly improved across all solutions (F (2,20) = 2.548; *p* = 0.540) however, significant effects of time (F (1.34,13.42) = 3.753; *p* = 0.01) and a solution × time interaction (F (2.55,25.47)= 1.132; *p* = 0.01) were observed. There was *a small* beneficial effect between MEN and PLA at 75–105 s (ES: 0.27; 90% CI: −0.45 to 0.96) as presented in Figure 2A.

Relative power (W/kg) was significantly affected by solution, (F (1.27,12.06) = 2.074; *p* = 0.05), time (F (1.40,14.03) = 4.049; *p* = 0.05) and further showed a significant solution × time interaction (F (2.62,26.25) = 0.811; *p* = 0.05), as per Figure 2B. *Small* differences were observed between solutions MEN and PLA (ES: 0.55; 90% CI:0.009 to 1.081), MEN and WAT (ES: 0.447; 90% CI: −0.085 to 0.59), through time periods 15–45 and 45–75 s. Solutions WAT and PLA saw *trivial* to *small* effects (ES: 0.429; 90% CI: −0.101 to 0.939) (ES: 0.274; 90% CI: −0.238 to 0.773) in time periods 75–105 and 135–165 s, respectively. Beneficial *moderate* differences between MEN and WAT (ES: 0.795; 90% CI: 0.204 to 1.352) and MEN and PLA (ES: 1.059; 90% CI: 0.412 to 1.666) were observed between 75–105 s, as well between 105–135 s when MEN was compared to both WAT (ES: 0.729; 90% CI: 0.152 to 1.276) and and PLA (ES: 0.791; 90% CI: 0.202 to 1.348). Further beneficial *moderate* effects were seen towards the end of the test, 135–165 s between solutions MEN and WAT (ES: 1.058; 90% CI: 0.411 to 1.665).

Differences in power output between conditions were further explored using thresholds for meaningful change (Figure 3), using fixed (12 W; Panel A) and funnelled approaches (12 W–5 W; Panel B). When compared to WAT, MEN exceeds meaningful change thresholds from 45 s onwards using the funnelled approach, and compared to PLA from 75 s onwards. These data are further supported by the effect statistics presented in Figure 4.

### 3.2. Physiological Responses

There were no significant solution × time interactions for absolute VO_2_ (F (3.652,36.523) = 0.937; *p* = 0.447), relative VO_2_ (F (2.605,26.050) = 0.347; *p* = 0.764), VE (F (3.130,31.301) = 0.293; *p* = 0.838) or RER (F (2.664,26.642) = 0.613; *p* = 0.594). Significant effects of time were however seen in all respiratory physiological variables (all *p* < 0.001). Furthermore, no significant effects were observed for solutions in absolute or relative VO_2_, VE and RER (*p* = 0.501 to 0.987). Blood Lactate (BLa) saw no significant effects for solution (F (2,20) = 0.462; *p* = 0.948), although there were significant effects for time (F (1.28,12.87) = 207.514; *p* = 0.01) and solution × time interaction (F (2.65,26.54) = 0.683; *p* = 0.024).

When inspecting pairwise comparisons, absolute and relative VO_2_ comparisons were *trivial* across all time points. VE was increased to a small extent following PLA rinsing, when compared to MEN, between 15–45 s (ES: 0.25; 90% CI: −0.45 to 0.96), all other comparisons were *trivial*. MEN induced *small* increases in RER at 15–45 s, 75–105 s, 105–135 s and 135–165 s when compared to PLA and WAT (Figure 5C), with the excepetion of a *moderate* increase when compared to PLA at 105–135 s (ES: 0.65; 90% CI: −0.07 to 1.37). All other comparisons were *trivial*.

No significant effects were observed for solution (F (2,20) = 1.099; *p* = 0.127) or time (F (4,40) = 361.461; *p* = 0.244) in relation to heart rate. A significant solution × time interaction was observed for heart rate (F (2.52,25.19) = 2.102; *p* = 0.04). Small effects were demonstrated between MEN and WAT during time points 0–90 s (ES: 0.30; 90% CI: −0.41 to 1.00), 90–180 (ES: 0.32; 90% CI: −0.39 to 1.02) and 180 s–recovery (ES: 0.43; 90% CI: −0.29 to 1.13). During the time period of 180 s–recovery small effects were also seen for solutions WAT vs. PLA (ES: 0.28; 90% CI: −0.43 to 0.98) and MEN vs. PLA (ES: 0.52; 90% CI: −0.21 to 1.21). Whilst temperature also showed no significant effect of solution (F (2.20) = 1.099; *p* = 0.118), there was a significant effect of time (F (1.37,13.67) = 10.388; *p* = 0.01) and solution × time interaction for temperature (F (2.43,24.27) = 1.125; *p* = 0.01).

### 3.3. Thermal Comfort, Thermal Sensation and RPE

There were no significant effects for solution (F (2,20) = 1.622; *p* = 0.014) and solution × time interaction for TC (F (3.15,31.53) = 0.664; *p* = 0.089). However, there were significant effects for time (F (4,40) = 54.473; *p* = 0.01). Between solution comparisons were all *unclear*, as illustrated in Figure 6A. Thermal sensation showed no significant effects for solution (F (2,20) = 0.577; *p* = 0.070) and solution × time interaction (F (8,80) = 1.689; *p* = 0.070). Significant effects were observed for time (F (1.13,11.33) = 4.264; *p* = 0.01). *Moderate* effects for thermal sensation were demonstrated between WAT and PLA (ES: 0.37; 90% CI: −0.39 to 1.02), MEN and PLA (ES: 0.35; 90% CI: −0.37 to 1.04) during time periods 90–180 s. *Moderate* effects also being observed during 180 s–recovery between WAT and PLA (ES: 0.54; 90% CI: −0.19 to 1.23), as presented in Figure 6B. No significant effects were observed in RPE for solution (F (2,20) = 0.873; *p* = 0.980). However, significant effects were demonstrated for time (F (1.71,17.16) = 0.225; *p* = 0.01) and solution × time interaction for RPE (F (3.75,37.50) =124.465; *p* = 0.014).

Small effects were seen across all three time points for RPE illustrated in Figure 6C. These were demonstrated during 0–90 s for MEN vs. WAT (ES: 0.24; 90% CI: −0.47 to 0.94) and MEN vs. PLA (ES: 0.34; 90% CI; −0.39 to 1.02). Also observed throughout 90–180 s for WAT compared to PLA (ES: 0.34; 90% CI: −0.38 to 1.04), MEN vs. PLA (ES: 0.25; 90% CI: −0.46 to 0.95) and 180 s–recovery for MEN compared to PLA (ES: 0.27; 90% CI: −0.45 to 0.96).

## 4. Discussion

The current study aimed to assess and compare the effects of menthol mouth rinsing (0.1%) on performance and thermal perception during a modified three-minute maximal test, to WAT and PLA rinses. Other rinses acted upon either the same oral receptors (WAT) or mimicked the flavour of MEN (PLA). The results demonstrated significant *small* to *moderate* differences in relative power output between solutions towards the end of the test (W/kg; Figure 2 and Figure 3; *p* = 0.05). Specifically, from 75–105 s between MEN and WAT (ES: 0.795; 90% CI: 0.204 to 1.352) and MEN and PLA (ES: 1.059; 90% CI: 0.412 to 1.666) this continued between MEN and WAT (ES: 0.729; 90% CI: 0.152 to 1.276) and MEN and PLA (ES: 0.791; 90% CI: 0.202 to 1.348) from 105–135 s. Between 135–165 s there was a *moderate* difference between solutions MEN and WAT (ES: 1.058; 90% CI: 0.411 to 1.665). Simply, this indicates that participants were producing more power for longer, having rinsed menthol prior to exercise, when compared to WAT or PLA rinsing. Practically, these findings suggest that when heat exposure is brief ≤ 15 min, participants may benefit from perceptual cooling despite not experiencing deleterious alterations in physiological temperature. Such heat exposures are most likely experienced by elite athletes who transition from air-conditioned or cool holding areas pre-competition, to hot competition arenas.

The outlined findings in test performance were supported by concomitant *small* to *moderate* increases in RER as a result of menthol rinsing, suggesting an increased glycolytic contribution to exercise and elevated bicarbonate buffering as a result of excessive CO_2_ production [59] may have contributed to greater relative power production. Such effects have previously been observed following carbohydrate rinsing [60] and supplementation [61] prior to exercise. Like carbohydrate rinsing, it has been suggested that menthol may elicit an increase in central drive and an increased sensation of pleasure albeit for a short period of time, potentially aiding performance [16,62]. Menthol is known to influence the drive to breathe and arousal through stimulation of the trigeminal nerves; when this occurs in hot environments this sensation is perceived as pleasant, which in turn may increases central drive and improves work capacity [16,23,24,25,62]. The absence of alterations in VO_2_ or VE between solutions is likely due to the maximal nature of the exercise task, eliciting relatively homogenous respiratory outcomes, independent of the solution administered. Stevens et al. [15] demonstrated *small* to *moderate* alterations in integrated electromyography (iEMG) activity as a result of physiological (30-min cold-water immersion at 23–24 °C plus ice ingestion), but not perceptual (MEN) cooling compared to control conditions in a 3 km time trial following a pre-load run. A combination of both strategies also induced iEMG changes. Performance enhancement relative to control conditions was deemed to have been driven by menthol administration (ES: 0.4; 90% CI: 0.2–0.6) and not as a result of increased iEMG activity, which was most affected by alterations in temperature. Menthol administration did induce a *small* increase in iEMG activity compared to control when administered immediately prior to time trial commencement, suggesting that whilst perceptual cooling may not confer as large or as sustained an elevation in iEMG activity as physiological cooling, statistically relevant increases in electrical activity may occur following menthol rinsing. Similar relationships between bitter tastes, pre-absorptive responses and exercise performance have also been observed [23,32,63].

Secondary findings from the current study demonstrated that there was little to no change in perceptual thermal and exertion measures as well as physiological ventilation measures, this potentially being due to the short duration of the heat exposure, the test itself as well as the intensity of the maximal style test [16]. Green [26] and Mündel and Jones [5] reported that orally administered menthol may make stimuli such as inspired air feel cool and thus increase sensations of nasal openness (patency; [17,24,25,26,27,62]). Menthol administration has been shown to elicit sensations of improved airflow without a change in airflow resistance, caused by the direct activation of the oropharyngeal cold receptors [62]. MEN did not significantly modify participants’ ventilation throughout the maximal test compared to WAT or PLA, but menthol has been postulated to reduce the effort of breathing [5], suggesting that if exercise were performed in domains of lesser intensity or was paced differently, alternate ventilatory responses may have been observed independent of other physiological factors. For instance, core temperature has been seen to increase at a steady rate throughout endurance trials, in proportion to either the work performed or the heat load experienced by the athlete (e.g., [35]). The outcome of aerobic exercise in the heat is individuals becoming fatigued more quickly, they have a decrease in cardiac output, higher glycogen use, higher core temperature, increased heart rate and increased rate of ventilation [36,37,64]. Heavy and prolonged exercise in the heat can result in hyperthermia, this occurring due to a state of exhaustion and reduction in stroke volume [36,37,65,66,67]. During the modified three-minute maximal test, there was no significant difference observed in core temperature and heart rate following MEN rinsing.

Exercising in the heat creates further stress on an athlete and during aerobic exercise in the heat individuals may become fatigued more quickly, as they have a decrease in cardiac output, higher glycogen utilisation, higher core temperature, increased heart and ventilatory rates [36,37,64,65,66]. An increase in muscle temperature may also be beneficial [67]. Despite a higher exercise intensity, because of the shorter exercise bout and thus heat exposure in the present study, participants are unlikely to have improved performance via increases in muscle temperature alone [67], as their exposure to an increased environmental temperature was only brief (≤15 min) and was not extreme. Increased muscle temperature is associated with increases in mechanical efficiency [36,67], but is susceptible to influence from muscle-fibre typology [67]. The hyperthermic potential of the exercise bout, and in some cases the athlete [36], need to be considered when translating the current findings into applied practice. If an athlete can thermoregulate effectively and the exercise duration is sufficiently short (e.g., ≤10 min), the exercise bout is unlikely to impart a meaningful heat dose for performance to be impaired, but the athlete may still benefit from menthol’s perceptual characteristics.

With regard to perceptual measures, neither thermal comfort nor sensation differed significantly between any of the three solutions. These results are not in agreement with the original hypothesis of the study, that there would be lowered perceptual measures, likely due to the short duration but high intensity exercise in the heat potentially limiting the hyperthermic potential of the exercise and the participants’ interpretation of thermal perceptions [36,37,68]. Gibson, et al. [43], reported improvements in perceptual measures with a study using menthol as a mouth rinse when undertaking repeated supramaximal sprints under laboratory conditions. Thermal comfort measures were improved after having menthol during exercise in comparison to capsaicin, placebo and control rinses. Although performance was not improved with the use of menthol, the value of perceptual alterations in heat studies involving menthol were reinforced, and thus indicated that further investigation was warranted. Expanding the use of menthol into shorter periods of exercise that exceed maximal aerobic capacity into the field could provide an easily administered ergogenic aid for athletes to use. Trialling menthol in multiple repeated supramaximal sprint tests or simulated team sport protocols, should be considered and may provide a wider range of performance and perceptual benefits due to the stop-start nature of the tests.

Riera et al. [10], also found no changes in perceptual measures (thermal comfort, thermal sensation and RPE) despite considerable enhancements in performance when menthol was added to beverages of varying temperatures. Menthol was added to neutral, cold and iced beverages and given to the athletes to ingest prior to a warmup, at the beginning of the test and every 5 km of a 20 km trial. Results showed no significant differences between beverages with all perceptual measures increasing over time. In a longer duration time trial over 40 km when MEN rinsing was compared to carbohydrate (CHO) or a MEN+ CHO rinse, similar findings were observed [35]. Menthol solutions did induce elevated localised sensations of oral cooling, but given extended exercise duration it was thought that glycogen reduction was a more deleterious signal than alterations in thermal perception as a result of exercise associated heat storage [35]. Gavel et al. [16], state that several previous studies have seen lower thermal perceptual measures however, there have been an equal number of studies reviewed that have observed no change at all, especially with increased exercise intensity. This is logical; despite administering an intervention that aims to lower thermal perception and RPE, with an increase in exercise performance (and presumably metabolic heat production due to ATP degradation) these measures may remain unchanged [3] or even worsen dependent upon the mechanisms by which performance enhancement is realised.

A classical model by Gagge et al. [68] demonstrates that differences in thermal comfort and sensation are not perfect correlates for increased core and skin temperatures, with perceptions of pleasure also considered a distinct input. Gagge et al. [68] also note that the verbal anchors used to describe sensations profoundly influence participant experience. Thus, ratings in the current study may have differed if participants were given a more thorough explanation of thermal comfort and sensation, where they should anchor their descriptions of each perception from and examples of when they might feel a particular number. For example, Best, et al. [30,31], used a scenario with the participants prior to testing to distinguish between thermal comfort and thermal sensation. The scenario consisted of participants imagining lying on a sun lounger in a hot environment (e.g., a beach). Thermal comfort being described as the degree of comfort experienced in this instance, whereas thermal sensation was described as the degree of heat stress perceived due to the environment (or exercise task). Participants may need to be repeatedly familiarised to be able to distinguish between perceptual states, in a low duration heat exposure, such as the present study. This would increase lab visits, experimental time and was deemed an inefficient use of resources, given the focus also being on exercise performance.

As the present study includes a maximal test, institutional ethical review did not permit the performance of multiple tests on the same day, despite evidence that performance (and recovery) may not be impaired [69]. However, repeated maximal performance (of similar durations) is a determinant of success in several individual and team sports formats, which may require athletes to contest a heat and a final, or multiple rounds of competition in one day e.g., athletics, cycling or rugby sevens. Therefore, a proposed extension to the line of inquiry established by the present work is that of either or both athlete durability (as per Maunder et al. [70]) and repeatability, where athletes’ decay in performance is assessed during prolonged exercise or with and without sufficient recovery between trials, respectively.

Pacing strategies amongst individuals during the tests differed despite all athletes having been familiarised and having had experience of maximal effort cycling on Wattbikes™ prior to study commencement. This is a limitation of the present study but is also an acknowledged limitation of the three-minute all-out test [40,41], with authors suggesting that if maximal exercise is not performed then the test should be repeated i.e., the test should be performed in a manner similar to the shorter duration Wingate test(s) [71]. Given the multiple conditions in the present trial, this would be taxing on resources and the participant, and may have actually reduced transferability of the present findings to applied competitive and training settings. Likewise, repetition of such trials may more closely mimic a conventional approach to ascertaining critical power, where time to fatigue at different power outputs is plotted as a hyperbolic decaying function [40,41,57,58], albeit with a narrower range of power outputs being assessed. As power output and not critical power was considered the performance variable of interest in the present study, the efforts given by our participants are sufficient to meet the aim of the study. We do acknowledge though that time to task failure, W’ and critical power are indeed interesting and may be modifiable as a result of oral menthol application or perhaps other tastants that may alter power output via non-absorptive pathways e.g., bitter tastes [32,72].

Intertest reliability is an important determinant of assessing change within an athlete as a result of a nutritional or training intervention, with pacing and performance levels being pertinent factors that are documented to influence three-minute all-out test performance [40,69,70]. Average power and end power obtained during a three-minute all-out test have been shown to be reliable between repeated trials [48,49], hence our focus on these metrics as performance thresholds for indicating meaningful change between interventions, as per Figure 2. We are satisfied with this interpretation, as an intervention is deemed meaningful if it exceeds the typical variation of the test, when compared to another condition. The time course of these effects is also of practical interest, given the hyperbolic relationship between power output and exercise duration(s). However, if we were to examine specific parameters underpinning power output, especially if exercise duration was extended, then these relatively low thresholds would have had to have been increased substantially and a greater degree of inherent variability accepted. McClave et al. [73], conducted a series of three-minute maximal tests with elite cyclists in a temperate environment and concluded that critical power may be overestimated during this test. Similar findings are provided by Bartram et al. [74] who indicate that the three-minute all-out test may overestimate critical power and underestimate W’ by 51 W (95% CI: 30 W–72 W) and −8.8 kJ (95% CI: −2.6 kJ to −15 kJ), respectively, compared to traditional multi-test duration profiling. Both McClave [69] and Bartram [74] report using elite cyclists, but there is a marked difference in mean VO_2max_ between participant groups of 59 mL/kg/min and 73 mL/kg/min, respectively, which may further influence the utility of the three-minute all-out test and explain some secondary outcomes of these investigations e.g., time to exhaustion at critical power.

Whilst we did not assess inter-test reliability within the present participant pool, due to the cost and health and safety implications of performing additional lab visits under pandemic protocols, researchers should aim to assess or at least understand sources of variability in their samples and thus aim to quantify or minimise them, whilst still presenting data in a transparent manner. We sought to minimise this within the present cohort by ensuring participants had multiple exposures to maximal exercise, testing order was assigned via randomisation and assessing effects via null-hypothesis significance testing, effect sizes and confidence intervals, in line with current recommendations [75,76]. These were further supported by fixed and funnelled statistical measures of variability from previously conducted research [57,58]. A future study may wish to examine variability of three-minute test performance in similar populations, especially given it being a more time effective measure of critical power and maximal oxygen uptake.

A further pertinent extension of the present work, which would incorporate elements of pacing and other applied and mechanistic factors would be to administer menthol to athletes who experience significant drafting effects e.g., as part of a team pursuit. Airflow appears critical in one’s experience of oropharyngeal sensations of cool and therefore menthol [35,62], thus if one were experiencing a significant drafting effect i.e., a reduction in wind resistance [77,78], the efficacy of menthol administration may be blunted but will still likely depend on other factors such as athlete preference and habituation to menthol and mentholated products, and TRPM8 receptor density [79,80,81].

## 5. Conclusions

The findings of this investigation suggest that 0.1% menthol used as a mouth rinse, prior to the completion of a modified three-minute maximal test in the heat, increases the relative power produced towards the end of the test, in comparison to a cold water or placebo mouth rinse. It is speculated that this is due to an increased glycolytic energy provision and increased bicarbonate buffering, as evidenced from an elevated RER, relative to WAT and PLA. However, no improvements in perceptual measures of thermal comfort, thermal sensation, RPE or ventilation during the test were observed—likely due to the duration of heat exposure and maximal nature of the exercise bout. Transitioning the use of menthol out of the laboratory and into the field may shed light on more appropriate means of using menthol as a tool to combat heat related performance decrements. Future research assessing menthol as a tool to combat heat related performance decrements in athletes should consider repeated short duration efforts (~3 min), critical power components and the potential role of airflow as a result of drafting behaviours.

## Figures and Tables

**Figure 1 ijerph-19-03527-f001:**
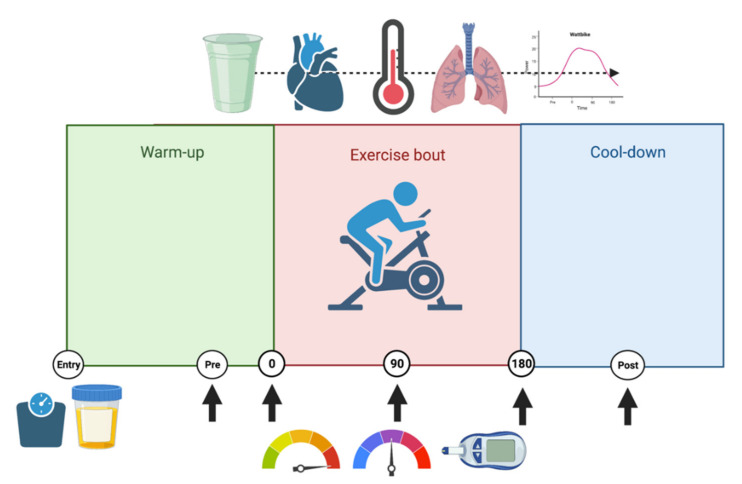
Experimental overview. Circles indicate time points as indicated in Methods and/or Results sections. Mass and hydration were assessed upon entry. The dashed horizontal arrow at the top of the image indicates continuous measurement at the sampling rates mentioned in the appropriate methodological subsections, from the point of menthol mouth-rinsing (green cup). Vertical arrows at the bottom of the image indicate measurement of rating of perceived exertion, thermal comfort and thermal sensation, and blood lactate. Created with Biorender.com (accessed on: 15 March 2022).

**Figure 2 ijerph-19-03527-f002:**
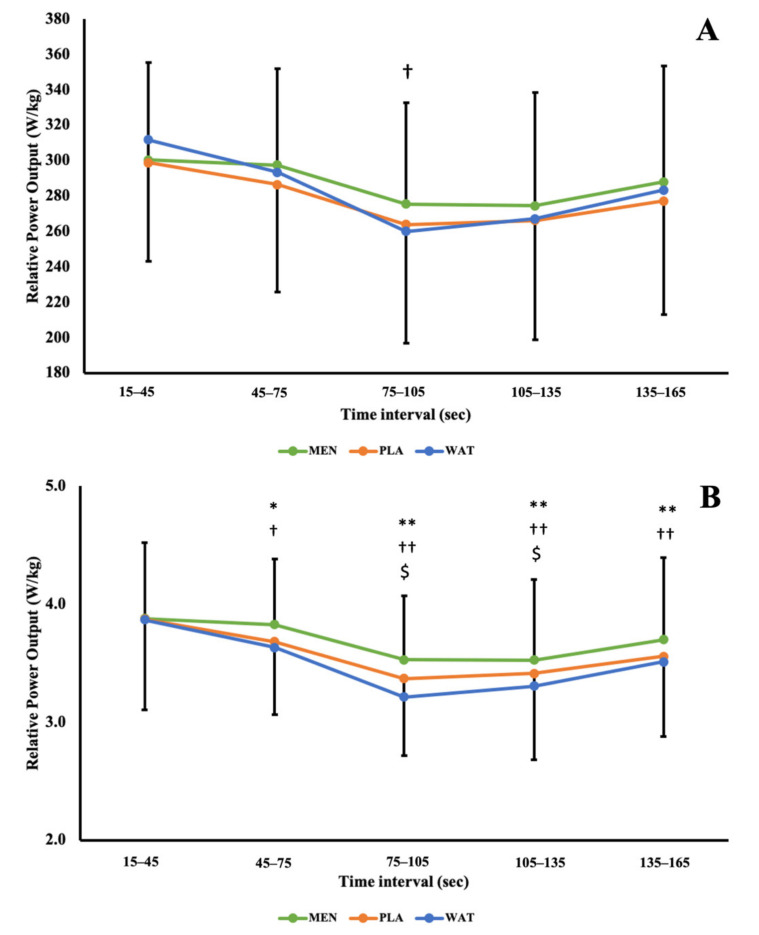
(**A**) Mean absolute power values (W) per condition. (**B**) Mean relative power values (W/kg) per condition. Error bars depict 1 SD; bidirectional error bars are not included per condition to maintain figure clarity. Symbols denote comparison (MEN vs. PLA *, MEN vs. WAT † and WAT vs. PLA $) and multiple symbols denote magnitude of effect (one, two and three of the same symbol denotes *small, moderate* and *large* effects, respectively).

**Figure 3 ijerph-19-03527-f003:**
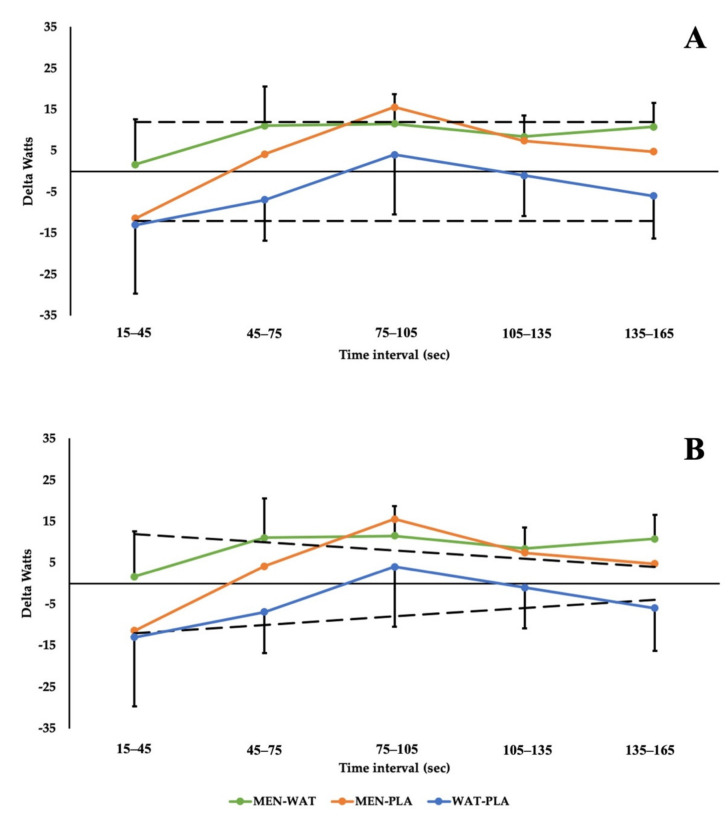
Differences in power output between conditions (delta W) and meaningful change. (**A**) considers meaningful change as ± fixed value (12 W); (**B**) considers meaningful change as a funnel, as time intervals progress, from 12 W to 5 W.

**Figure 4 ijerph-19-03527-f004:**
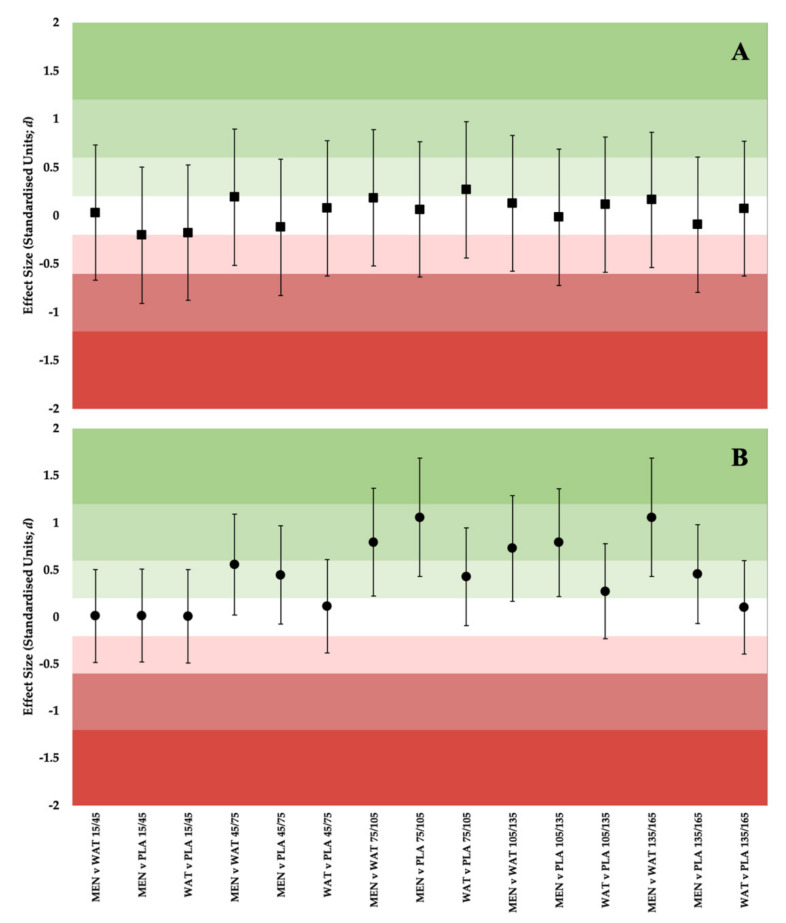
Forest Plot showing ES ± 90% CI at matched time points between solutions for (**A**) Absolute Power; (**B**) Relative Power.

**Figure 5 ijerph-19-03527-f005:**
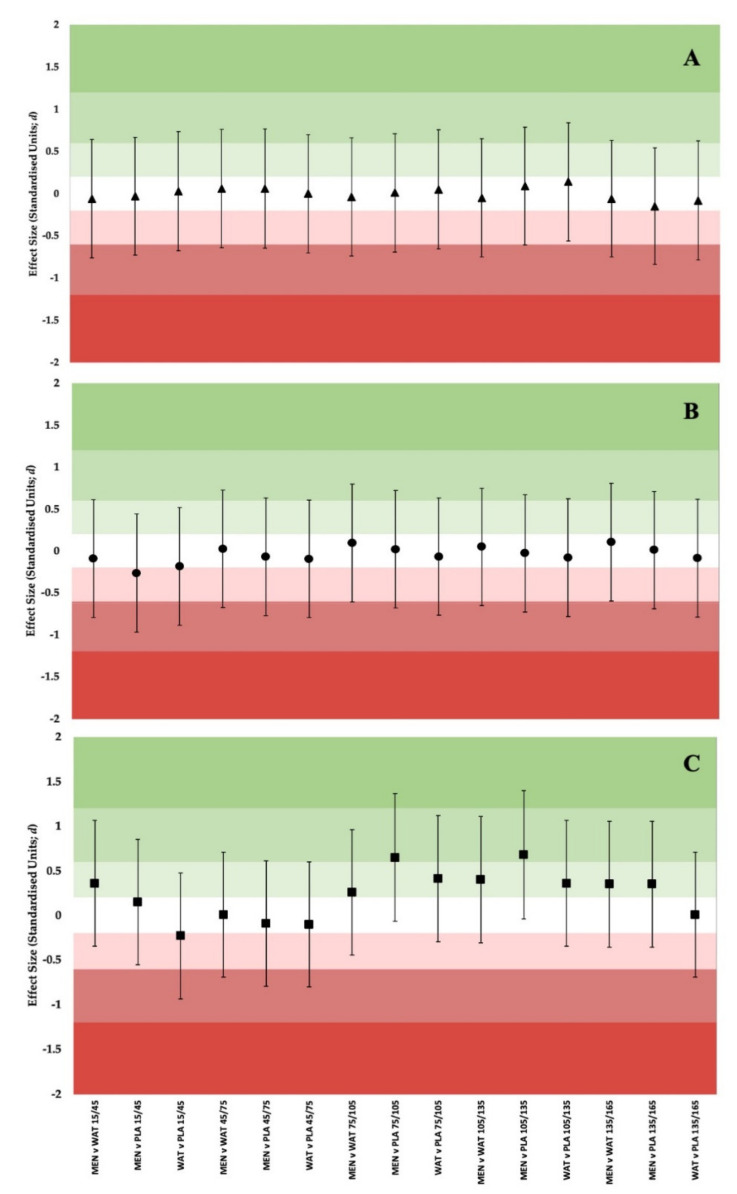
Forest Plot of ES ± 90% CI at matched time points and between solutions for (**A**) VO_2_,(**B**) VE and (**C**) RER.

**Figure 6 ijerph-19-03527-f006:**
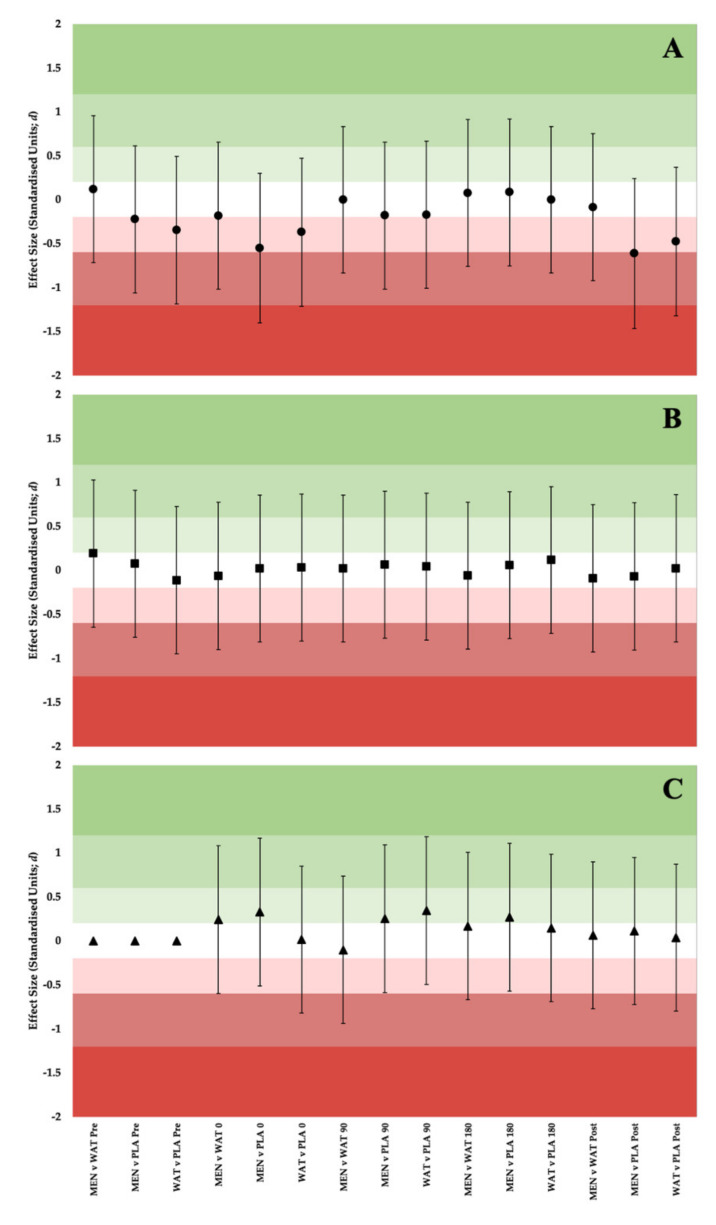
Forest Plot of ES ± 90% CI at matched time points and between solutions for (**A**) Thermal Comfort, (**B**) Thermal Sensation and (**C**) Rating of perceived exertion.

## Data Availability

The data presented in this study are available on request from the corresponding author. The data are not publicly available due to institutional data policies, however an anonymised, secure copy will be stored on Researchgate.

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
