# Peer review of "Menthol Mouth Rinsing Maintains Relative Power Production during Three-Minute Maximal Cycling Performance in the Heat Compared to Cold Water and Placebo Rinsing"

_ijerph, 2022, doi:10.3390/ijerph19063527_

Round 1

Reviewer 1 Report

The authors explored the effect of 3 mouth rinses (PLA, cold water, and MEN) on 3-minute maximal cycling performance in a hot environment. Please find attached my specific comments to the manuscript. The methods were done fairly well apart from minimum familiarization to a test that has a known learning effect and the lack of determining each individual's test-re-test reliability . The results section needs to be improved and the conclusions softened in light of the small difference in power with MEN compared to PLA and CON. 

Author Response

Thank you for kindly reviewing our paper, please find attached a response to reviewers' comments and an amended manuscript.

Reviewer 2 Report

There was no strong justification to propose that Menthol Mouth rinse may potentially be of benefit to short 3 min maximal exercise. The explanation provided was based on studies that employed endurance exercise.

Detail comments are provided in the manuscript as attached.

Author Response

(The authors gave the same response as above.)
